# Electroanalysis Applied to Compatibility and Stability Assays of Drugs: Carvedilol Study Case

**DOI:** 10.3390/ph13040070

**Published:** 2020-04-17

**Authors:** Murilo Ferreira de Carvalho, Luane Ferreira Garcia, Isaac Yves Lopes de Macedo, Ricardo Neves Marreto, Mayk Teles de Oliveira, Renê Oliveira do Couto, Carlos Eduardo Peixoto da Cunha, Karla Carneiro de Siqueira Leite, Kênnia Rocha Rezende, Fabio Bahls Machado, Vernon Somerset, Eric de Souza Gil

**Affiliations:** 1Faculty of Pharmacy, Federal University of Goiás, Goiânia 74605-170, Brazil; murilo.fc@hotmail.com (M.F.d.C.); luane.fg@hotmail.com (L.F.G.); isaacyvesl@gmail.com (I.Y.L.d.M.); rnmarreto@gmail.com (R.N.M.); mayk.teles@hotmail.com (M.T.d.O.); cedcunhap35@hotmail.com (C.E.P.d.C.); kennia@gmail.com (K.R.R.); famafarm@yahoo.com.br (F.B.M.); 2Pharmacy School, Federal University of São João del-Rei, Midwest Campus, Divinópolis–MG, ZIP 35.501-296, Brazil; rocouto@ufsj.edu.br; 3Universidade Paulista, Goiânia, GO 74845-090, Brazil; karlacsl11@gmail.com; 4Department of Chemistry, Faculty of Applied Sciences, Cape Peninsula University of Technology, Bellville 7535, South Africa; vsomerset@gmail.com

**Keywords:** anti-hypertensives, drug delivery systems, electrochemical impedance spectroscopy, electrochemistry, excipients, pharmaceutical technology

## Abstract

Carvedilol (CRV) is a non-selective blocker of α and β adrenergic receptors, which has been extensively used for the treatment of hypertension and congestive heart failure. Owing to its poor biopharmaceutical properties, CRV has been incorporated into different types of drug delivery systems and this necessitates the importance of investigating their compatibility and stability. In this sense, we have investigated the applicability of several electroanalytical tools to assess CRV compatibility with lipid excipients. Voltammetric and electrochemical impedance spectroscopy techniques were used to evaluate the redox behavior of CRV and lipid excipients. Results showed that Plurol^®^ isostearic, liquid excipient, and stearic acid presented the greatest anode peak potential variation, and these were considered suitable excipients for CRV formulation. CRV showed the highest stability at room temperature and at 50 °C when mixed with stearic acid (7% w/w). The results also provided evidence that electrochemical methods might be feasible to complement standard stability/compatibility studies related to redox reactions.

## 1. Introduction

Carvedilol (CRV) (Figure 1) is a non-selective blocker of α and β adrenergic receptors, and has been extensively used for the treatment of hypertension and congestive heart failure. CRV oral bioavailability ranges from 20% to 35% owing to its low water solubility and increased fast metabolism [1,2,3]. To circumvent its unsatisfactory biopharmaceutical properties, CRV has been incorporated in different types of drug delivery systems, e.g., nanoemulsions [4], solid-state cyclodextrin complexes [5], solid dispersions [6], transdermal vehicles [7], and supramolecular hydrogels [8].

A considerable number of lipid-based drug delivery systems have been developed for overcoming CRV pharmacokinetic issues [4,9,10]. Even though CRV stability and compatibility studies with solid pharmaceutical excipients have been reported in the literature [11,12,13], only a few investigations dealing with CRV stability in lipid vehicles have been performed so far [14].

Thermal and spectroscopic methods have been widely used in drug–excipient compatibility studies [15]. Differential scanning calorimetry (DSC), differential thermal analysis (DTA), and Fourier-transform infrared spectroscopy (FTIR) are amongst the most popular techniques. These techniques require a small amount of sample, and can provide a rapid analysis with high sensitivity. However, there are several drawbacks related to these techniques, such as the high cost of equipment acquisition and maintenance, fastidious sample preparation, difficult data interpretation (DSC and DTA) and interference of environmental variables, such as humidity (FTIR) [15].

Due to the relevance of compatibility studies and the limitations of the usual analytical techniques, several researchers have been performing studies for the development of reliable and low-cost approaches to investigate the compatibility/stability of pharmaceutical products [16,17,18]. Henceforth, electrochemical methods might be useful when compatibility studies may be driven by redox reactions. As long as the oxidation of chemicals is retarded or hastened by the environment, electrochemical methods have a great potential for the study of drug–excipient compatibility. These techniques can provide significant insights concerning thermodynamic and kinetic features related to degradation rates. 

Moreover, electrochemical methods are noteworthy for their affordability and environmentally-friendly advantages [19,20]. Given that most excipients are non-electroactive, the usefulness of electrochemical approaches for the analytical determination of drugs in pharmaceutical formulations is well stated [19,20,21,22]. Notwithstanding, electrode reactions are frequently coupled by chemical reactions, in which the sheer versatility of electroanalysis to explore the thermodynamic and kinetic aspects of the redox process, could be applied to bring some highlights in compatibility studies. 

Given the importance that CRV displays in clinical practice as well as the relevance of developing innovative tools to study drug–excipient compatibility, we report on the applicability of the electroanalytical tools for assessing the CRV compatibility with lipid excipients.

## 2. Results

### 2.1. Electrochemical Characterization of Carbon-Paste Electrodes (CPEs)

The cyclic voltammetry (CV) scans containing different agglutinating systems, as described in Table 1, are presented in Figure 2. The system containing the lipophilic excipient, Plurol^®^ isostearic, presented two oxidation peaks (Figure 2A), whose peak intensity increased accordingly to its concentration in carbon paste (CP), thus indicating the inherent electroactivity. On the other hand, the CP prepared with vegetable oils and oleic and stearic acids did not present any anodic peak (Figure 2B; data from fatty acid analysis are not shown). On the other hand, the Capmul carbon paste (CP^Cap^) swelled when immersed in the PBS solution, even when Capmul concentration was lowered to 0.5%. Owing to this physical instability, no further assays were conducted with this excipient.

### 2.2. Electrochemical Evaluation of CRV and Binary Systems at Solid State

The evaluation of CRV compatibility with some non-electroactive lipid excipients was performed by dissolution or dispersion of 1% of the drug in the agglutinating system described in Table 1. The differential pulse voltammograms obtained for the CP control and some modified CPs with and without CRV are presented in Figure 3. It can be observed that the anodic peak potential (*E*_pa_) and the anodic peak current (*I*_pa_) values of CRV shifted in all cases, when compared to CRV in the CP control. In turn, Compritol^®^ and stearic acid produced the opposite effect on CRV electrochemical parameters, *E*_pa_ and *I*_pa_, (Figure 3B).

For the results obtained for Plurol^®^ isostearic, that exhibited electroactivity in CV scans (Figure 2A), it was shown that its anodic peaks (dotted line, Figure 3C) occurred at distinct *E*_pa_ values of CRV (grey line, Figure 3C). Meanwhile, the binary system similar to stearic acid produced the largest shift in the peak currents’ values (double dotted dashed line, Figure 3C).

Thence, the thermodynamic parameter, *E*_pa_, would indicate that the excipients herein investigated lead to an anodic shift for CRV. Table 2 presents the displacements of first anodic peak potentials (Δ*E*_p1a_*,* of electrochemical parameters, *E*_p1a_ and *I*_p1a_, obtained for 1% CRV in the CP control and modified CPEs. The liquid excipients used were oleic acid (OA), sesame oil (SeO), canola oil (CO), safflower oil (SfO), and Plurol^®^ (Pl). The solid excipients were Emulium^®^22 (Emul), Compritol^®^ (Cpt); and estearic acid (EA).

From the positive values of Δ*E*_p1a_, it can be inferred that the over potential required for CRV to undergo electrochemical oxidation is greater in the presence of lipophilic excipients. Beyond that, a lower tendency to undergo oxidative chemical reaction would also be expected. The greater *I*_pa_ values may be attributed to greater dissolution of CRV or to enhancement of electron transfer kinetics in the presence of the excipient. Both aspects would favor chemical reaction. Moreover, stearic acid, Compritol^®^, Plurol^®^, and safflower carbon pastes showed statistical difference (ANOVA and paired Student’s T-test).

Hence, together with the electroactive character, modifiers can alter the electrodic properties, thus leading to noticeable changes for *E*_pa_ and *I*_pa_, of the CPs (Table 1), when submitted to electrochemical impedance spectroscopy (EIS) study using the potassium ferri/ferrocyanide system as redox probe (Figure 4).

As shown in Figure 4, the CV scans obtained for ferri/ferrocyanide probe were slightly distorted when the Nujol^®^ fraction of CP was modified with 20% of Plurol^®^ isostearic. This fact is attributed to increasing resistivity of CP material, being evidenced in the Nyquist plot, in which the arc for CPE^Pl20%^ is far higher than the one observed for the CPE control (Figure 4 and Table 1).

An enhancement of solution resistance (*R_s_*) and charge transfer resistance (*R_ct_*) was observed for other CPEs. The related EIS data of Randles circuit parameters are presented in Table 3. It is possible to assert that each lipophilic agglutinant altered the conducting and capacitive properties of electrodes in different ways, which could have to shifting of both *E*_pa_ and *I*_pa_ parameters.

### 2.3. Effect of Time and Temperature

Aiming to evaluate the impact of lipophilic excipients on CRV chemical stability, different CPEs were studied as a function of time at room temperature (25 ± 2 °C; Figure 5), and under conditions of forced thermal degradation (Figure 6). In this study, the target modified CPs were the ones that presented greater Δ*E*_p1a_ values. 

Each binary system was stored at room temperature (25 ± 2 °C) and under darkness from 0 to 180 days, whereas the drug decay was checked over this timeframe, utilizing DPV analysis. Also, some samples were stressed at 50 °C to evaluate the effect of temperature in combination with compatibility.

It is possible to see in Figure 5 that CRV showcased suitable stability even after 180 days of storage at room temperature (25 ± 2 °C). The variance of the degradation curves (Figure 5) was tested utilizing chi-squared test (Appendix A). No variances showed any significant statistical difference (probability of null hypothesis, that variances are equal, greater than the significance level α 0.05), except for Plurol^®^ isostearic. This effect may be attributed to the lack of solubility of CRV; thus, the tested oils showed a similar degradation profile.

The calibration graph obtained at solid state for increasing amounts of CRV in the CP control as well as in each modified CP, allowed DPV decay determination. The greatest stability was achieved for stearic acid, in which more than 90% of CRV was recovered. The results obtained are in agreement with the greatest value of over potential, in which the anodic shift, Δ*E*_p1a_ = 0.418 V, for stearic acid (Table 3) is also the greatest.

To evaluate the effect of higher temperatures on the CRV decay in CP binary systems, forced degradation studies were performed. The target samples were CPEs containing binary mixtures of 1% CRV and safflower 30%, Plurol^®^ isostearic 20%, or stearic acid 7% as well as CPs prepared with crushed tablets of CRV. Results are displayed in Figure 6. 

It can be observed in Figure 6A that CPEs containing the crushed tablet were more stable than pure CRV dispersed in Nujol^®^, the PC control. This fact might relate with stabilizing agents present in tablet. In Figure 6B, it was observed an initial peak current increase for CRV in CP^PI20%^, which can be attributed to inner dissolution of the active compound. The other CP mixtures showed a similar behavior compared to the CRV in CP control.

## 3. Discussion

The electrochemical characterization of CPEs prepared with different proportions (10–30% for liquids, 1.5–7% for solids) of the distinct lipid excipients, was performed in order to determine their solid-state electrochemical behavior. This procedure is necessary to avoid mistakes on the drug–excipient compatibility evaluation. For instance, for electroactive excipients, the inherent anodic peaks must be prior known, whereas in the case of non-electroactive species, the resistivity increasing of the agglutinating system could also lead to misunderstandings. Likewise, the electrochemically inert compounds would be new alternatives of agglutinating systems to prepare pseudo-CP electrodes.

The lack of electroactive character, as well as, the negligible impact on the conducting properties in comparison to CP control (Table 1), enable such excipients to be used as alternative agglutinant oils.

This fact can be attributed to the formation of an emulsified system able to disturb the graphite matrix of the CP. Indeed, the surfactant character may also explain the increment observed for capacitive current in CP^PI30%^ (Figure 2A and Figure 3C).

Plurol^®^ is a polyglycerol ester with HLB (hydrophilic lipophilic balance) of ca. 9, conferring to this compound, the higher surfactant properties among the target lipophilic excipients, herein investigated [22]. The surfactant property, the electron donor character of the long aliphatic chains becomes the unsaturated bonds and hydroxyl groups oxidizable at lower potentials, thus explaining the anodic peaks, 1a and 2a (Figure 2A and Figure 3). 

Due to the complete saturation of the aliphatic chain, the mineral oil (Nujol^®^), is expected to be the most inert agglutination system.

Therefore, the EIS and voltammetric characterization of modified CPEs is mandatory to avoid misinterpretations concerning the impact of each excipient on the CRV oxidation. Therefore, the thermodynamic (*E*_pa_) and kinetic (*I*_pa_) electrochemical parameters may be evaluated taking this fact into account. 

The CRV increase observed for CP^PI20%^ (Figure 6B) may be attributed to the solubilization process of the drug in this binary system, resulting from the highest temperature and from the surfactant properties of Plurol^®^ isostearic. Meanwhile, the greater the molecular dispersion, the greater the vulnerability to undergo chemical reactions. This may explain the further decay observed after 3 days of storage.

According to Silva et al. [14], CRV is compatible with stearic acid, Emulium^®^, Plurol^®^ isostearic, and Compritol^®^. These authors used analytical techniques such as DTA, DSC, FTIR, and HPLC analysis to draw these conclusions, which are in agreement with data from cyclic voltammetry measurements. 

Additionally, the electrochemical analysis indicated that stearic acid and Plurol^®^ isostearic lead to a higher oxidation potential for CRV and, as a consequence, they have a higher protective effect against CRV oxidation. 

Silva et al. [14] also reported that Capmul^®^ MCM and oleic acid are incompatible with CRV. Similar findings were reported in the present study, since oleic acid showed the lowest potential shift associated with the highest oxidation current peak, indicating that this excipient makes the CRV more prone to be oxidized. Canola and safflower oils are composed of a great amount of oleic acid and showed similar findings. In turn, sesame oil has in its composition an appreciable extent of antioxidants (sesamin, sesamolin, and sesaminol), which may confer higher oxidative stability. Together, the electrochemical results gathered in this study are in agreement with previously- published data, indicating the potential use of the electrochemical method to investigate drug–excipient compatibility, thus expanding the applications of solid-state voltammetry [23]. 

## 4. Materials and Methods

### 4.1. Reagents and Solutions

Carvedilol (CRV), 99.27% purity, was supplied by Chengtai Shenyang Fine Chemical Factory (China). Graphite, mineral oil (Nujol^®^) and stearic acid were purchased from Sigma-Aldrich (Saint Louis, MO, USA). Oleic acid was purchased from Labsynth LTDA (Brazil). Compritol^®^ 888 ATO (glyceryl behenate, GB), Plurol^®^ isostearic (polyglyceryl-6-isostearate), Emulium^®^22 (tribehenin PEG-20 esters) and Capmul^®^ MCM (glyceryl caprylate/caprate) were kindly donated by Gattefossé (France). Canola oil was provided by Cargil Agrícola S.A. (Brazil), safflower oil was purchased from Pazze Indústria de Alimentos LTDA (Brazil), and sesame oil from Croda (Brazil). 

Potassium ferrocyanide, K_4_[Fe(CN)_6_].3H_2_O and potassium phosphate salts (analytical grade), were purchased from Vetec Química Fina Ltd. (Rio de Janeiro, Brazil) and diluted in purified water in order to reach a final concentration of 0.05 mol L^−1^. Thereafter, KCl was added to this solution up to a concentration of 0.1 mol L^−1^. The 0.1 mol L^−1^ phosphate buffer solution (pH 7.0) was prepared with ultra-pure Milli-Q water (Millipore S.A. Molsheim, France, conductivity ≤ 0.1 μS cm^−1^). 

### 4.2. Electrochemical Assays of Stability and Compatibility

#### 4.2.1. Voltammetric Assays

Voltammetric measurements were performed using a potentiostat/galvanostat PGSTAT^®^ model 204 with module FRA32M (Metrohm Autolab) integrated with NOVA 2.1^®^ software. The measurements were performed in a 25 mL one-compartment electrochemical cell, with a three-electrode system consisting of carbon-paste electrodes (CPEs) described in Table 1, and Pt wire and Ag/AgCl/KCl_sat_ (both purchased from Lab solutions, São Paulo, Brazil) representing the working, counter, and reference electrode, respectively. The carbon paste was mechanically renewed with every new analysis performed.

The experimental conditions for differential pulse voltammetry (DPV) were as follows: pulse amplitude = 25 mV, pulse width = 0.5 s, and scan rate = 10 mV s^−1^. The experimental conditions for cyclic voltammetry (CV) were as follows: scan rate = 50 mV s^−1^ and scan range from −0.4 to 1.4 V. The DPV were background-subtracted and baseline-corrected, and then all data were analyzed and treated with the software Origin 8^®^. All experiments were performed at room temperature (25 ± 2 °C) in triplicate (*n = 3*) and the main electrolyte used was the 0.1 mol L^−1^ KCl, 0.1 mol L^−1^ phosphate buffer solution (PBS) at pH 7.0.

#### 4.2.2. EIS Characterization

Electrochemical impedance spectroscopy (EIS) measurements were conducted in a solution containing 0.05 mol L^−1^ potassium ferrocyanide in 0.1 mol L^−1^ KCl over a frequency ranging from 0.1 Hz to 100 kHz at selected potentials for sensors.

### 4.3. Electrochemical Evaluation of Oxidative Stability of Lipophilic Compounds 

The vulnerability to undergo spontaneous oxidation is inwardly related to anodic peak potential, *E*_pa_, which was obtained utilizing solid-state voltammetry. This can be implied due to the relationship between Gibbs free energy and the electric potential, as stated by several authors [23]. In this context, the target compounds, drugs, and selected lipophilic excipients were incorporated as modifiers in carbon-paste electrodes (CPE). The carbon pastes (CP) were composed of graphite powder and mineral oil or other modified agglutinating system that was composed by using different proportions of the target samples (Table 1). 

The liquid CP modifiers were rigorously mixed with graphite powder by spatulation technique. The solid modifiers were homogenized before dispersion in the mineral oil fraction. In the next step, the CP mixtures (Table 1) were inserted in the supporting CP electrode (E), which consisted in Teflon tube centrally pierced by 2 mm diameter copper wire, to leave a cavity of 2 mm diameter and 0.5 mm depth.

The resulting CPEs were immersed in buffer solutions for 5 min before electrochemical assays, to allow CP conditioning. In these studies, the anodic peak current, *I*_pa_ (kinetic parameter) and anodic peak potential, *E*_pa_, (thermodynamic parameter) of each component, as well as their effect on the electrode conductivity, were carefully evaluated utilizing voltammetric and EIS techniques. 

#### 4.3.1. Electrochemical Evaluation of Redox Compatibility 

The effect of each excipient on the drug redox stability was evaluated at CPE, in which 1 mg of CRV was homogeneously dispersed into binary agglutinating systems (Table 1), before mixing with graphite powder. The drug thermodynamic tendency to suffer redox reactions (electron donor ability) is expressed by the negative shifting of the anodic peak. In turn, the electron transfer kinetics, when the amount of electroactive species is kept constant, is expressed by peak current values. The impact of each lipophilic excipient on CRV redox behavior expresses their compatibility and was evaluated in the light of such electrochemical parameters.

##### Effect of Time and Temperature

The stability of binary mixtures of CRV and different lipid excipients (Table 1), was evaluated utilizing DPV, in the function of time at room temperature (25 ± 2 °C). The effect of isothermal stress was also evaluated by submitting each CP binary system to 50 °C for 10 days. All necessary controls were evaluated.

### 4.4. Statistical Analysis

To evaluate the analytical data, statistical tests consisting of chi-square (for variance), paired Student’s T-test, and ANOVA (for average) were performed. All statistical analysis was performed in Origin 9.0^®^ software package (Northampton, MA, USA). The confidence interval was set to 0.95 and statistical significance was attributed to *p* < 0.05.

## 5. Conclusions

The electroanalytical techniques, such as cyclic and differential pulse voltammetry, at solid-state, showed to be useful tools to evaluate the drug–excipient compatibility of lipophilic compounds. Moreover, the aforementioned analytical methods can be extended to predict drug stability against oxidative reactions. In this context, it was also found that some vegetable oils could replace mineral oil as an agglutinating system. Furthermore, among the lipophilic excipients, Plurol^®^ isostearic and estearic acid, presented the greatest values of Δ*E*_p1a_, corroborating with stability studies in binary systems obtained by other methods. Therefore, electroanalysis can also be explored as a complementary tool in compatibility and stability studies, especially when redox process is strongly related to chemical degradation.

## Figures and Tables

**Figure 1 pharmaceuticals-13-00070-f001:**
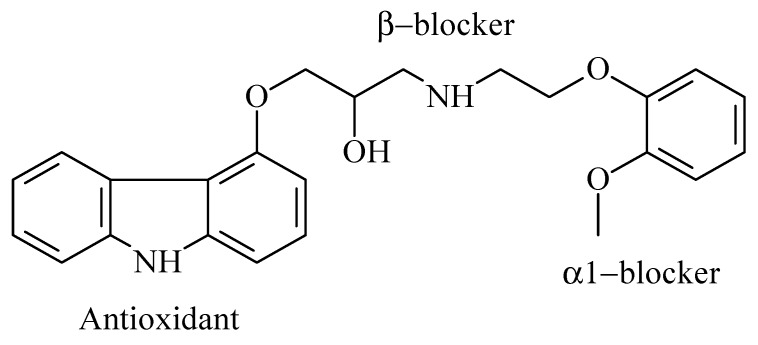
Chemical structure of carvedilol.

**Figure 2 pharmaceuticals-13-00070-f002:**
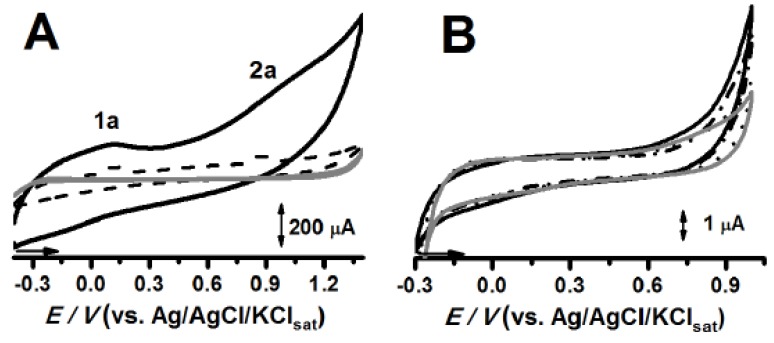
Cyclic voltammograms of CP control (─), CP^PI20%^ (---), and CP^PI30%^ (─) (**A**); CP^SfO30%^ (─), CP^SeO30%^ (•••), and CP^CO30%^ (-•-•-) e CP control (─) (**B**). All experiments were performed in triplicate in 0.1 mol L^−1^ KCl, PBS, pH 7.0. Plurol^®^ isostearic carbon paste (CP^PI^), safflower oil carbon paste (CP^SfO^), sesame oil carbon paste (CP^SeO^), canola oil carbon paste (CP^CO^).

**Figure 3 pharmaceuticals-13-00070-f003:**
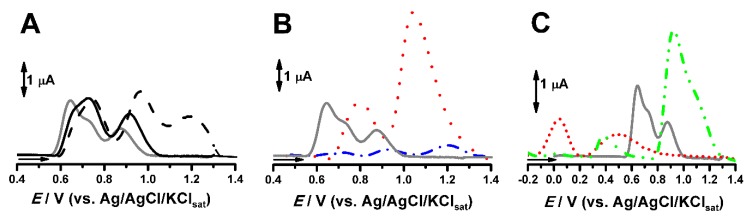
Solid-state differential pulse voltammograms of 1% CRV in: CP control (grey line of Figure 3A–C) and in CP^SeO30%^ (—) and CP^CO30%^ (- - -) (**A**), CP^Cpt7%^ (- • - •) and CP^EA7%^ (• • •) (**B**), CP^PI20%^ without CRV (• • •) and with 1% CRV (- • • -) (**C**). All assays were performed in 0.1 mol L^−1^ KCl, PBS, pH 7.0. Sesame oil carbon paste (CP^SeO^), canola oil carbon paste (CP^CO^); Compritol^®^ carbon paste (CP^Cpt^); estearic acid carbon paste (CP^SfO^); Plurol^®^ isostearic carbon paste (CP^Pl^) and Carvedilol (CRV).

**Figure 4 pharmaceuticals-13-00070-f004:**
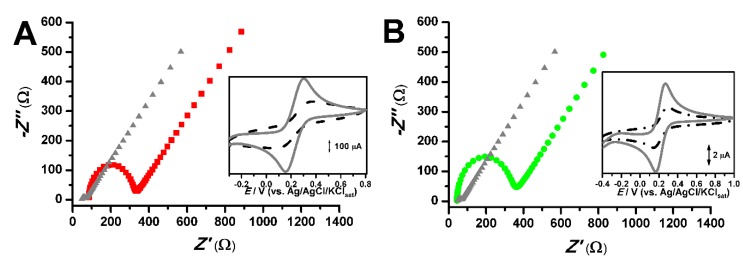
Electrochemical impedance Nyquist plots of CP control (■). (**A**) CP^PI20%^ (●) and (**B**) CP^EA7%^ (●). Insert: the related cyclic voltammetry (CV) scans. CP control (──), CP^PI20%^ (- - -), and CP^EA7%^ (- • -). All CV and electrochemical impedance spectroscopy (EIS) assays were performed with 0.05 mol.L^−1^ potassium ferri/ferrocyanide in 0.1 mol L^−1^ KCl solution.

**Figure 5 pharmaceuticals-13-00070-f005:**
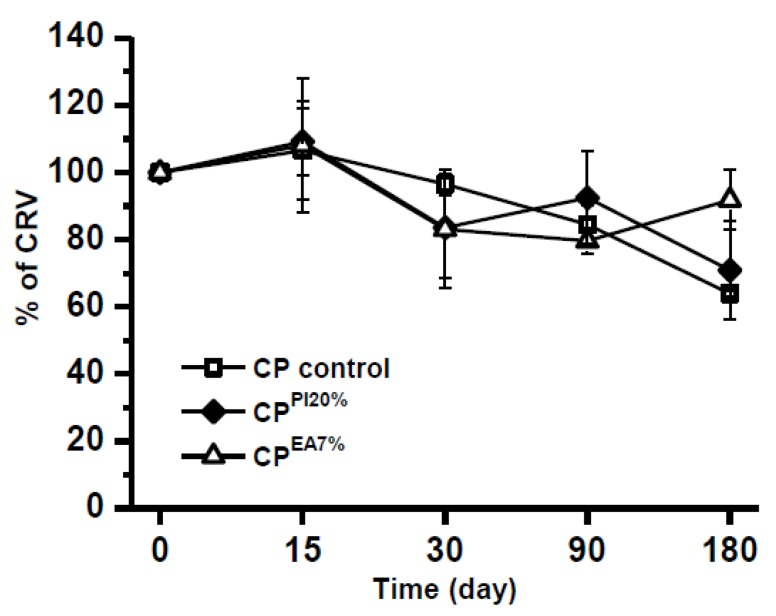
Results shown for CRV decay in different CPs stored at room temperature for up to 180 days.

**Figure 6 pharmaceuticals-13-00070-f006:**
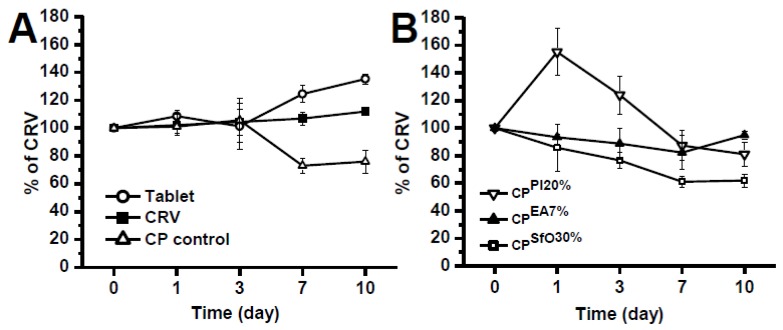
Results shown for CRV decay in different CPs binary systems at a 50 °C (**A**) and under darkness for up to 10 days of storage (**B**).

**Table 1 pharmaceuticals-13-00070-t001:** Carbon paste compositions for stability assays.

Lipophilic Modifiers	(mg)	Mineral Oil (mg)	Graphite Powder (mg)
Liquid excipients ^1^	10	20	70
	15	15	70
	20	10	70
	30	-	70
Solid excipients ^2^	1.5	30	68.5
	3	30	67
	7	30	63
Carvedilol (CRV) control	1	30	70
Carbon Paste (CP) control	-	30	70

^1^ Oleic acid (CP^OA^), Capmul^®^ (CP^Cap^), Plurol^®^ isostearic (CP^PI^), safflower (CP^SfO^), canola oil (CP^CO^) or sesame oil (CP^SeO^); ^2^ stearic acid (CP^EA^), Emulium^®^22 (CP^Emu^) or Compritol^®^ 888 ATO (CP^Cpt^).

**Table 2 pharmaceuticals-13-00070-t002:** *E*_p1a_ and *I*_p1a_ values obtained for 1% (m/m) CRV in the carbon paste electrode (CPE) control and different modified CPEs.

Excipients	*E*_p1a_ (V)	*I*_p1a_ (µA)	Δ*E*_p1a_ (*E*_p1a_–Cp Control *E*_p1a_)
**CP control**	0.625 ± 0.025	1.881 ± 0.285	---
**Liquid Excipients**			
CP^OA^	0.670 ± 0.004	5.679 ± 0.283	0.045
CP^SeO^	0.689 ± 0.062	1.937 ± 0.236	0.064
CP^CO^	0.706 ± 0.048	2.013 ± 0.490	0.081
^ѣ^ CP^SfO^	0.727 ± 0.006	4.089 ± 0.179	0.102
^ѣ^ CP^PI^	0.919 ± 0.001	3.105 ± 0.523	0.294
**Solid Excipients**			
CP^Emul^	0.660 ± 0.001	3.410 ± 2.405	0.035
^ѣ^ CP^Cpt^	0.930 ± 0.010	0.205 ± 0.056	0.305
^ѣ^ CP^EA^	1.043 ± 0.015	4.850 ± 1.816	0.418

^ѣ^ One-way ANOVA and paired Student’s T-test—probability of the null hypothesis (means are equal) smaller than the significance level (α = 0.05).

**Table 3 pharmaceuticals-13-00070-t003:** Randles equivalent circuit elements for each electrode system evaluated.

Circuit Elements	Nujol^®^	CPE ^Pl20%^	CPE ^EA7%^
*R_s_*	48.78 Ω	85.25 Ω	44.79 Ω
*R_ct_*	17.31 Ω	232.75 Ω	291.5 Ω
*C*	5.95 µF	1.89 µF	1.04 µF
*Y*	5.63 mMho.s^1/2^	3.94 mMho.s^1/2^	1.82 mMho.s^1/2^

Solution resistance (*R_s_*); charge transfer resistance (*R_ct_*); pseudo-capacitance of the system (*C*); admittance (*Y*).

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
