# Peer review of "Electroanalysis Applied to Compatibility and Stability Assays of Drugs: Carvedilol Study Case"

_pharmaceuticals, 2020, doi:10.3390/ph13040070_

Round 1

Reviewer 1 Report

The authors present electroanalytical data on the activity and Carvedilol (CRV), an alpha and beta adrenaline receptor blocker, and discuss their findings on how the electroanalytical response relates to CRV solubility in various excipients. Data is clearly presented and a thorough Materials and Methods section explains how experiments were performed. The paper is well written, well motivated, and presents and confirms useful information on the electrochemical properties of Carvedilol in various lipid excipients. The authors do an excellent job incorporating the possibility of temperature-induced degradation and studying how that may alter CRV redox behavior.

I suggest a few very minor revisions, but otherwise find the work to be suitable for publication in MDPI Pharmaceuticals.

-- Please add a clear and concise Conclusions section.

-- Figures are rendered with low quality, somewhat pixelated graphics. Please revise figures to be higher resolution. Additionally, figures would benefit from being rendered in color. Also, why is Figure 2A cut off on the left side of the image? Please revise to include all data collected for Figure 2A.

-- Figure 3B and 3C: The depicted lines on the graphs do not match the lines on described in the figure text. Please correct this to make it easier for the reader to match CPxxx values with lines on the graphs.

-- Similar to above, please clarify Figure 4, possibly also adding color.

Best regards.

Author Response

Response:

The conclusion section has been updated and a concise version of it was added.

-- Figures are rendered with low quality, somewhat pixelated graphics. Please revise figures to be higher resolution. Additionally, figures would benefit from being rendered in color. Also, why is Figure 2A cut off on the left side of the image? Please revise to include all data collected for Figure 2A.

Response:

The figures of low quality were corrected and updated.

Figure 2A was not cut, but since the voltammogram started at – 0.40 V, the beginning of the scan may seem torn.

-- Figure 3B and 3C: The depicted lines on the graphs do not match the lines on described in the figure text. Please correct this to make it easier for the reader to match CPxxx values with lines on the graphs.

Response:

The captions were updated and color was added to make it more descriptive and easier to read.

-- Similar to above, please clarify Figure 4, possibly also adding color.

Response:

For Fig. 4, the captions were updated and color was added to make it more descriptive and easier to read.

The authors thanks a lot the reviewer

Reviewer 2 Report

Electroanalysis applied to compatibility and stability assays of drugs: Carvedilol study case

This manuscript could be publishable after major revision.

  • The abbreviation must be explained at the beginning, such as CPEs, CP, DP, CPPI20%, CPPI30%, CPSfO30%, CPSeO30%, CPCO30….
  • The experimental procedures have not been clearly explained.
  • There are some grammatical, linguistic and spelling mistakes in the manuscript and will need to be addressed.
  • Figures should be formatted properly.
  • In Figure 2, the anodic peak obtained in the Cyclic voltammograms of CP control can behave differently in different pH. I think the pH needs to be evaluated for this work.
  • The effect of time and temperature should be explained in more details.

Author Response

  • The abbreviation must be explained at the beginning, such as CPEs, CP, DP, CPPI20%, CPPI30%, CPSfO30%,CPSeO30%, CPCO30….

Response:

An explanation of the abbreviation was provided.

  • The experimental procedures have not been clearly explained.

Response:

Sections of the methodology and procedure were updated to clearly outline the experimental procedure.

  • There are some grammatical, linguistic and spelling mistakes in the manuscript and will need to be addressed.

Response:

The manuscript was subjected to grammarly.com and all grammar related issues were corrected.

  • Figures should be formatted properly.
  • In Figure 2, the anodic peak obtained in the Cyclic voltammograms of CP control can behave differently in different pH. I think the pH needs to be evaluated for this work.

Response:

We acknowledge the referee’s remarks. However, pH is known to alter ΔEp only in electroactive compounds. In this work, CP contained mineral oil, which is known to exhibit electrochemical stability upon larger potential windows than the ones herein used, which supports its use as control.

Considering that pH might only influence thermodynamics and kinetics in electroactive compounds, we considered the development of this work solely under pH 7.0, so that it would not impair data reproducibility.

  • The effect of time and temperature should be explained in more details.

Response:

We develop these studies to showcase the correlation between thermal and oxidative degradation, and how these phenomena can be evaluated using low cost approaches such as electrochemistry.

In this sense, assays were performed in solid state to investigate the correlation between electrochemical thermodynamic stress and temperature increase.

Round 2

Reviewer 2 Report

This manuscript is publishable now.